# Increasing Performances of 1–3 Piezocomposite Ultrasonic Transducer by Alternating Current Poling Method

**DOI:** 10.3390/mi13101715

**Published:** 2022-10-11

**Authors:** Ke Zhu, Jinpeng Ma, Yang Liu, Bingzhong Shen, Da Huo, Yixiao Yang, Xudong Qi, Enwei Sun, Rui Zhang

**Affiliations:** 1Department of Physics, Harbin Institute of Technology, Harbin 150080, China; 2School of Instrumentation Science and Engineering, Functional Materials and Acousto-Optic Instruments Laboratory, Harbin Institute of Technology, Harbin 150080, China; 3Key Laboratory for Photonic and Electronic Bandgap Materials, School of Physics and Electronic Engineering, Harbin Normal University, Harbin 150025, China

**Keywords:** ultrasonic transducer, alternating current polarizing, 1–3 piezocomposites, ultrasonic imaging

## Abstract

Ultrasonic transducers are the basic core component of diagnostic imaging devices, wherein the piezoelectric materials are the active element of transducers. Recent studies showed that the alternating current poling (ACP) method could develop the properties of piezocomposites, which had great potential to improve transducer performance. Herein, transducers (*f*_c_ = 3 MHz) made of DCP and ACP 1–3 piezocomposites (prepared by PZT-5H ceramics and PMN-PT single crystals) were fabricated. The effect of the ACP method on the bandwidth and insertion loss (sensitivity) was explored. The results indicate that the ACP method can significantly enhance the bandwidth and slightly increase the insertion loss of transducers. Particularly, a superhigh bandwidth of 142.8% was achieved in the transducer of ACP 1–3 PMN-PT single crystal combined with suitable matching and backing layers. This bandwidth is higher than that of all reported transducers with similar center frequency. Moreover, the optimization mechanism of transducer performance by the ACP method was discussed. The obtained results suggested that the ACP is an effective and convenient technology to improve transducer performances, especially for the bandwidth.

## 1. Introduction

In recent years, high-performance ultrasonic transducers have been widely used in many industrial and scientific areas, including energy conversion, non-destructive determination, information acquisition, medical diagnosis, etc. [1,2,3]. Especially for ultrasound-based diagnostic imaging with the advantages of being efficient, nonradiative and operating in real-time, ultrasonic transducers are usually applied for visualizing internal body structures [4]. Generally, a shorter pulse length will incur a better axial imaging resolution in diagnostic-imaging transducers, wherein a shorter pulse length can be achieved by increasing the operating frequency or enhancing the bandwidth of transducers [5]. However, the high operating frequency will result in a large acoustic attenuation, further decreasing the detecting range of transducers; thus, the operating frequency of diagnostic-imaging transducers is below 10 MHz and is usually fixed at 3–5 MHz to ensure a reasonable detecting range [6,7]. In this case, ultrasonic transducers with broad bandwidth are always preferred. Moreover, it is generally known that sensitivity is another essential technical parameter for ultrasonic transducers [8]. However, fabricating ultrasonic transducers with both broad bandwidth and high sensitivity remains technically challenging [2,8].

In an ultrasound transducer, the piezoelectric layer is the most crucial component as the active element to achieve the energy conversion between mechanical energy and electrical energy [9]. Thus, the electromechanical properties of piezoelectric materials mainly determine the performance parameters of transducers [10]. It summarizes that a large piezoelectric coefficient (*d*) can increase the sensitivity, and a high electromechanical coupling factor (*k*) contributes to the broad bandwidth [11]. At present, the piezoelectric materials used in transducers include piezoelectric ceramics, single crystals and composites [12,13,14]. Among them, Pb (Zr_1−x_Ti_x_)O_3_ (PZT) ceramics and Pb(Mg_1/3_Nb_2/3_)O_3_-PbTiO_3_ (PMN-PT) single crystals are the most widely applied in transducers owing to their excellent electromechanical properties and outstanding performance stability [15,16]. For example, the PMN-PT single crystals with morphotropic phase boundary (MPB) composition present the excellent electromechanical properties of *d*_33_ > 2000 pC/N, *k*_33_ > 90% and high-electric-field strain *S* > 1.5% [17]. Considering the acoustic impedance mismatch between piezoelectric materials and biological tissue, the 1–3 piezocomposites based on PZT ceramics and PMN-PT single crystals are preferred to fabricate the medical ultrasonic transducers [18,19,20]. In addition, it is reported that the 1–3 piezocomposites have a higher *k* value than piezoelectric bulk materials of the same type, which benefits a broader bandwidth in 1–3 piezocomposite transducers [19,20]. Albeit ultrasound transducers have been widely applied in diagnostic imaging areas, further improving the transducer performances to meet the needs of coming high-performance medical diagnostic devices with more precision and resolution is always an attractive research topic [21]. Focusing on this topic, exploring an effective and simple method to improve the *d* and *k* of 1–3 piezocomposites while maintaining low acoustic impedance is of vital importance.

Several methods have been proposed to improve the electric properties of 1–3 piezocomposites, such as structure design, volume fraction regulation and stress engineering [21,22,23]. However, there are very limited ways to improve the electromechanical properties while maintaining low acoustic impedance. Several works have recently confirmed that alternating current poling (ACP) effectively improves the dielectric and electromechanical properties of ferroelectric single crystals and ceramics [24,25,26]. The optimization mechanism of the ACP method is usually associated with domain structure tailoring. It is generally recognized that the uniform domain structure combined with a small domain size afforded the performance enhancement in ACP samples [27]. Our previous works further confirmed the applicability of the ACP method in improving the electromechanical properties, including the *d* and *k*, especially in the PMN-PT-based 1–3 piezocomposites [28,29]. Meanwhile, the low acoustic impedance can be maintained in ACP 1–3 piezocomposites. These positive effects make the ACP a potential and advantageous technology to improve the transducer performances. Although the application prospect is anticipated, the actual effect of the ACP method on transducer performances is still not determined.

In this work, the ultrasonic transducers with the center frequency of 3 MHz were fabricated using the 1–3 piezocomposites. The effect of the ACP method on the dielectric and electromechanical properties of 1–3 piezocomposites was studied, as in Figure 1a [30,31]. Further, the effect of the ACP method on the bandwidth and insertion loss (sensitivity) of transducers was clarified, as in Figure 1b,c. The optimization mechanism of transducer performance by the ACP method was discussed.

## 2. Materials and Methods

PMN-30PT single crystals (Shanghai Institute of Ceramics, Shanghai, China) were chosen as the piezoelectric element to prepare 1–3 piezocomposites by the cutting and filling method. Commercial PZT-5H piezoelectric ceramics (Baoding Hongsheng Acoustical Electronics Inc., Baoding, China) were selected as comparison. The epoxy resin (Epotek301, Technology, Inc., Billerica, MA, USA) served as the flexible filler. As the electrodes, Au was sputtered on the opposite surfaces of prepared samples. The DCP samples were poled at the DC high voltage to implement the performance comparison. In the ACP process, the Function/Arbitrary Waveform Generator (Agilent) was applied to generate a symmetrical triangular wave. Then, the AC voltage was amplified by a high-voltage amplifier (Trek Model10/40A). Both ACP and DCP samples were poled at room temperature in silicone oil. The free dielectric constant ($\varepsilon_{33}^{T}{/\varepsilon }_{0}$) was measured by the Agilent 4294A multi-frequency LCR meter at 1 kHz. The electromechanical coupling coefficient (*k*_t_) was calculated by the impedance resonance method according to the IEEE standard. The piezoelectric coefficient (*d*_33_) was attained using a quasi-static ZJ-2 Piezo *d*_33_ meter.

In the preparation procedure of matching and backing layers, the filler powders (100-nanometre alumina (Al_2_O_3_) particles, 5-micron lead (Pb) particles and 5-micron tungsten (W) particles) were, respectively, added to the resin matrix while stirring frequently; then, the mixture was placed in a vacuum chamber for 10 min to remove the bubbles. After standing the mixture for 24 h to solidify the resin, the samples were to implement the cutting and polishing process. The matching layer and backing with various filler powder in the volume ratio range from 0% to 30% were prepared. To fabricate the designed transducers, the prepared piezoelectric composites, matching and backing layers were cut and polished into desired thickness. The matching and backing layers with the required thickness were pasted to both sides of the piezoelectric element using an epoxy resin adhesive, then fixing the copper wires and encapsulating the transducer. Pulse-echo experiment was used to characterize fabricated transducer performance. All ultrasonic transducers were excited by an electrical impulse of 2 μJ and a reputation rate of 1 kHz using an ultrasonic pulse source (5072 PR, Olympus) wherein damping = 50 Ω and gain = 0, and the pulse-echo signal was collected by an oscilloscope. The transducer was controlled by an electric translation platform, moving 0.1 mm each time and 500 steps in total.

## 3. Result and Discussions

### 3.1. Preparation of Piezocomposites

In this work, the PZT-5H ceramics and PMN-30PT single crystals were first immerged in silicone oil of 300 degrees for 1 h to ensure complete depolarization. Next, they were crafted into 1–3 composite structures. The designed piezocomposite structure is illustrated in Figure 2a, wherein the piezoelectric element was cut into cubes with the size of 0.63 (thickness) × 0.18 × 0.18 mm^2^, and the gap was 0.07 mm. The suitable size ratio (0.18/0.63 ≈ 0.28) of piezoelectric rods and appropriate volume ratio (≈52%) of the piezoelectric phase can ensure the pure electromechanical coupling mode. Meanwhile, this structure design contributes to achieving a high electromechanical coupling coefficient (*k*) in 1–3 piezocomposites. The realistic picture of prepared PZT-based piezocomposites is given in Figure 2b,c to show the surface micrograph of PZT-based piezocomposites. It is observed that the sample has clear boundaries and a closely periodic microstructure.

### 3.2. Poling Investigation

It is generally known that poling process is crucial for the ferroelectric single crystals and ceramics to perform the electromechanical conversion capability. As an effective performance optimization method, alternating current poling (ACP) has been widely authenticated in ferroelectric materials [25,26,27,28,29,30]. This section focuses on the effect of the ACP method on the dielectric and electromechanical properties of 1–3 piezocomposites to guide further performance optimization of the transducer.

Based on systematic procedures, the ACP conditions, including poling electric field amplitude (*E*_p_), poling cycle number (*C*) and poling frequency (*f*) of 1–3 piezocomposites, were studied, respectively. First, the samples were completely depolarized. Then the *E*_p_ increased from 0 to 3 kV/mm, the *C* increased from 1 to 20, and the *f* increased from 0 to 20 Hz, respectively. Ten samples were measured for each poling condition, and their average was used for the comparison. The optimal AC polarization conditions of 1–3 piezoelectric composites were finally found and summarized in Table 1. The DCP PZT and PMN-PT piezocomposites were poled at 2 kV/mm and 1 kV/mm for 30 min, respectively, which can well ensure that the DC polarization reaches saturation state.

Figure 3a,d show the iron electromagnetic hysteresis (*P*–*E*) curves of ACP and DCP samples under an electric field of 25 kV/cm at room temperature. As can be seen that the blue curve representing ACP is wider and higher, which means after the ferroelectric material was polarized and external electric field was removed, the polarization strength (*P*_r_) maintained was larger. Larger residual polarization means that the material has better piezoelectric properties, so the preparation of piezoelectric materials requires the *P*_r_ value to be as large as possible. Conversely, the piezoelectric performance of the sample can be reflected by the test of the *P*_r_ value of the sample. According to the *P–E* hysteresis loop, the *P*_r_ value of the AC-polarized sample is larger than that of the DC-polarized one, which is strong evidence that the ACP method can improve the performance of piezoelectric materials. We have found similar patterns in several experiments, and our previous studies have shown similar results [28,29]. In addition, some related research reports are also consistent with our results [32,33,34,35].

In the test in Figure 3, samples were obtained under optimal AC and DC polarization conditions, which ensured that both methods reached the optimal effect.

Figure 3c–f show the impedance spectra of 1–3 piezocomposites under different polarization conditions. It can be seen from the impedance spectrum that the resonant and anti-resonant peaks of AC polarization are further away than those of DC polarization. The electromechanical coupling coefficient *k*_t_ is calculated by the following formula:(1)$$k_{t}=\sqrt{1-\frac{f_{r}^{2}}{f_{a}^{2}}}$$
where *f*_r_ and *f*_a_ represent resonant and antiresonant frequencies.

Through calculation, it is clearly found that the electromechanical coupling coefficient *k*_t_ of ACP sample is higher than that of DCP sample, indicating that transducers made from ACP sample will have better bandwidth.

The specific test results are shown in Table 2. For PZT-based piezocomposites, an increase of 16.8% for $\varepsilon_{33}^{T}{/\varepsilon }_{0}$, 29.6% for *d*_33_ and 13.6% for *k*_t_ are observed in ACP samples compared to the DCP samples. Moreover, there is an enhancement of 18.6% for $\varepsilon_{33}^{T}{/\varepsilon }_{0}$, 14.05% for *d*_33_ and 20.5% for *k*_t_ in ACP PMN-PT-based piezocomposites. It should be noted that a high *k* value could improve the bandwidth of the transducers; meanwhile, a larger *d* value benefits achieving a higher sensitivity. Thus, the performance improvement in ACP samples is expected to implement high-performance transducer technology.

### 3.3. Preparation of Transducers

The additional acoustic layers, including the front matching layer and backing layer, also play an important role in improving the transducer performance. The matching layer is applied to minimize the acoustic impedance mismatch between piezoelectric materials (~30 MRayls) and biological tissue or water (~1.5 MRayls) to avoid acoustic energy loss at the interface. The backing layer is used to deliver the reflected wave and absorb part of the energy from the vibration of the back face while acting as a supporting layer of the fragile piezoelectric element. Generally, the metal-loaded resin is an ideal choice to take the role of acoustic layers owing to their proper acoustic impedance and attenuation. In this work, we explore the acoustic properties of Al_2_O_3_-, Pb-, and W-loaded resin to select suitable acoustic matching materials. Figure 4a gives the variation in acoustic impedance (*Z*) of metal-loaded resin as a function of the volume ratio of Al_2_O_3_, Pb and W. It is observed that all the *Z* levels indicate a monotonous rising trend with the increasing metal-powders volume. At the same metal-powders volume ratio, the W-loaded resin has the maximum *Z* value while the Al_2_O_3_-loaded resin has the minimum value; the *Z* value of Pb-loaded resin is between the two parties. The average size of tungsten and lead particles is 5 microns, while the average size of alumina is 100 nanometers, so the sound attenuation of Al_2_O_3_-loaded resin will be significantly less than the former two. The results above suggest that the W-loaded resin is a suitable backing layer material because of its high acoustic impedance and large attenuation. The large acoustic attenuation can accelerate the energy dissipation of reflected waves in the backing layer, and the high acoustic impedance of the backing layer is instrumental in the broad bandwidth of the transducer. Moreover, the Al_2_O_3_-loaded resin is the proper matching layer material because its low acoustic attenuation can avoid acoustic loss, while suitable acoustic impedance can ensure acoustic transmission efficiency in matching layers.

Generally, the matching layer thickness is set to *λ*/4, wherein *λ* is the wavelength in the matching layer material. The two matching layers method can sometimes be applied to improve the acoustic matching structure further, and their acoustic impedances were chosen to be 6.5 MRayls and 3.8 MRayls, which can well improve the acoustic impedance mismatch between piezocomposite and load.

The influence of backing acoustic impedance on transducer bandwidth and insertion loss (simulated by PiezoCAD) is shown in Figure 4b. It can be seen that, with the increase in backing acoustic impedance, transducer bandwidth increases significantly, while insertion loss decreases slowly. These two parameters cannot be maximized at the same time. Here, we choose to prioritize the high bandwidth of the transducer at the expense of a little sensitivity. The tungsten particles were compressed to the limit by centrifugation to obtain the densest backing with the largest acoustic impedance (11.5 MRayls). The designed transducer structure is illustrated in Figure 4c, and the photograph of fabricated transducers is given in Figure 4d. The physical dimension and acoustics property parameters of piezoelectric elements, matching and backing layers are summarized in Table 3. The designed center frequencies of transducers are the same as 3.0 MHz.

### 3.4. Performances of Fabricated Transducers

The insertion loss and bandwidth are critical performance parameters for transducers, which determine the imaging quality for the greater part. In this work, the performances of the fabricated transducer were measured using a conventional pulse-echo response measurement method. The two-way insertion loss (*IL*) is calculated by Equation (2) as follows:(2)$$IL=20\log\left(\frac{V_{0}}{V_{i}}\right)$$
where *V*_0_ and *V*_i_ are the output and input voltages of the transducer, respectively. The *IL* (or the relative pulse-echo sensitivity) is the ratio of the transducer output voltages to the input voltages under the condition that output resistance and input resistance are approximately equal. Utilizing the built-in Fourier-transform (FFT) feature of the oscilloscope, the frequency spectrum of the pulse-echo response was determined. The center frequency (*f*_c_) and −6 dB bandwidth (*BW*) of the transducer can be calculated from the acquired FFT spectrum:(3)$$f_{c}=\frac{1}{2}\left(f_{1}+f_{2}\right)$$
(4)$$BW=\frac{2\left(f_{2}-f_{1}\right)}{f_{1}+f_{2}}\times 100\%$$
where *f*_1_ and *f*_2_ refer to the lower and upper −6 dB frequencies in the spectrum, respectively.

Figure 5 gives the pulse-echo response and FFT spectrum of fabricated transducers by DCP and ACP methods. The center frequencies of all the transducers are around the expected 3.0 MHz. The transducer of ACP PZT ceramic has a −6 dB bandwidth of 112.1% and an insertion loss of −17.6 dB; by contrast, there is an increase of 22.3% for bandwidth and 1.2 dB for insertion loss than that of DCP one (BW = 89.8% and IL = −18.8 dB). Meanwhile, the −6 dB bandwidth and insertion loss of the transducer using ACP PMN-PT single crystal reaches 142.8% and −16.1 dB, respectively, which is 34.9% and 0.9 dB higher than the DCP one. These results indicate that the ACP effectively improves the bandwidth and insertion loss of the transducer simultaneously. The technical parameters of fabricated transducers by DCP and ACP methods are summarized in Table 4.

Figure 6 shows the scan graph of the fabricated transducers for the wire box model. It is observed that the reflected signals are clear when the transducer sweeps across the section of the copper wires. The enlarged images of the signal in the middle are also given in Figure 6. In each enlarged image, one can see that the signal closer to the probe is stronger with a wider transverse range. As expected, the signals obtained by the ACP transducers (both the transducers of PZT-5H ceramic and PMN-PT single crystal) are more obvious and brighter, while the image depth of each copper wire signal is smaller. This phenomenon has resulted from the improvement of *d*_33_ and *k*_33_ in ACP samples. In addition, the transducers of PMN-PT single crystals present a higher image resolution than the transducers of PZT-5H ceramics. It is due to the general knowledge of better electromechanical performances of PMN-PT single crystals than PZT-5H ceramics. The specific axial and lateral values of −6 dB resolution are shown in Table 5.

Generally, the bandwidth of the transducer is mainly determined by the *k* value of the piezoelectric element; that is, a larger *k* incurs a broader bandwidth. Herein, obtaining superhigh bandwidth can be summarized as the following factors: (i) the 1–3 piezocomposite structure brings a higher *k* value than that of bulk piezoelectric elements, (ii) the ACP method further improves the *k* value for 1–3 piezocomposites; especially, the *k* (0.88) of 1–3 PMN-PT single crystal piezocomposite is very excellent in reported piezocomposites, and (iii) suitable matching and backing layers contribute to the enhancement of transducer bandwidth. These multiple cooperations incur the superhigh bandwidth transducer in this work. In addition, a large *d*_33_ value in ACP piezoelectric elements ensures an excellent sensitivity in fabricated transducers. Accordingly, the ACP method achieved the simultaneous optimization of bandwidth and sensitivity in the transducers.

## 4. Conclusions

In this work, the superhigh bandwidth transducers with excellent sensitivity were fabricated by the mean of ACP improving the electromechanical properties of the piezoelectric element. The effect of ACP on 1–3 piezocomposites based on PZT-5H ceramics and PMN-PT single crystals was studied. The results showed that the dielectric, piezoelectric and electromechanical properties of ACP piezocomposites are significantly improved compared with those of DCP samples. An increase of 29.6% for $\varepsilon_{33}^{T}{/\varepsilon }_{0}$, 16.8% for *d*_33_ and 13.6% for *k*_33_ were achieved in ACP PZT-5H ceramic piezocomposite, while an increase of 14.1%, 18.6% and 20.5% for $\varepsilon_{33}^{T}{/\varepsilon }_{0}$, *d*_33_ and *k*_33_ were observed, respectively, in ACP PMN-PT single crystal piezocomposite. Then, the transducers with various piezoelectric elements (*f*_c_ = 3 MHz) were fabricated combined with suitable matching and backing layers, and the performances of different transducers were compared. It suggested that the ACP method could simultaneously improve the bandwidth and insertion loss of the transducers. The bandwidth and insertion loss improvement originates from increased *k*_33_ and *d*_33_ under ACP technology. Particularly, a superhigh bandwidth of 142.8% was acquired in the transducer of ACP PMN-PT single crystal, which is higher than that of all reported transducers with similar center frequency. The contributing factors of ultrahigh bandwidth are the collective effect of the ACP technology and suitable matching and backing layers. Our works provide an effective strategy for the collaborative optimization of the bandwidth and sensitivity of transducers, further guiding the design of high-performance ultrasonic transducers used in medical diagnosis.

## Figures and Tables

**Figure 1 micromachines-13-01715-f001:**
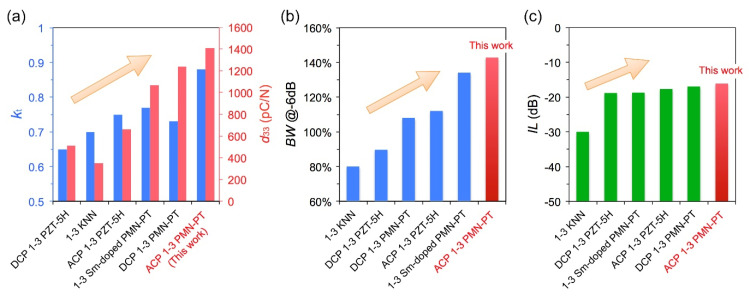
The (**a**) *k*_t_ and *d*_33_ factor of different 1–3 piezocomposites. The (**b**) −6 dB bandwidth and (**c**) two-way insertion loss of transducers with different 1–3 piezocomposites.

**Figure 2 micromachines-13-01715-f002:**
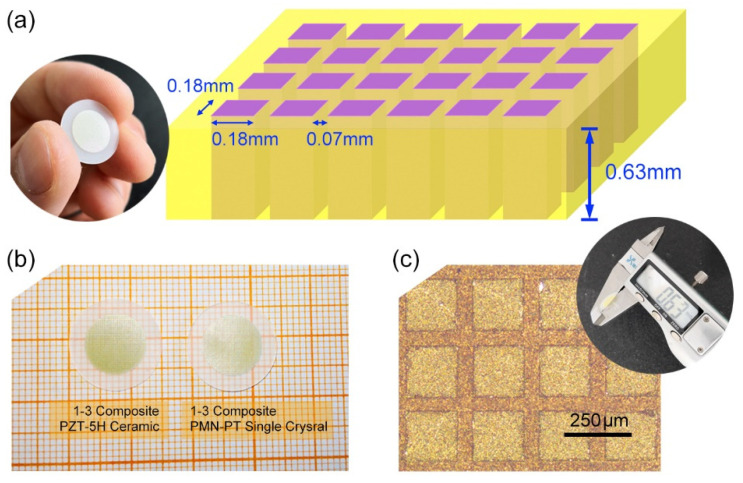
(**a**) Schematic of 1–3 piezoelectric composites, (**b**) photographs of PZT-based 1–3 piezocomposite, (**c**) surface micrograph of PZT-based 1–3 piezocomposites.

**Figure 3 micromachines-13-01715-f003:**
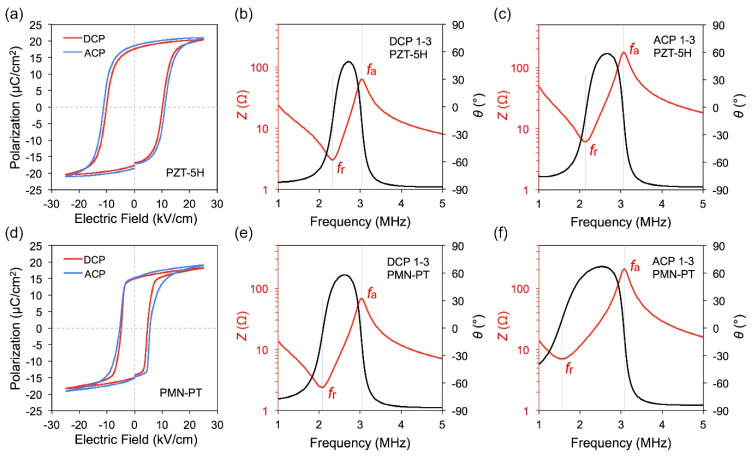
The iron electromagnetic hysteresis curves of (**a**) 1–3 PZT-5H and (**d**) 1–3 PMN-PT. The impedance spectrums of (**b**) DCP 1–3 PZT-5H, (**c**) ACP 1–3 PZT-5H, (**e**) DCP 1–3 PMN-PT and (**f**) ACP 1–3 PMN-PT acquired by Agilent 4294A.

**Figure 4 micromachines-13-01715-f004:**
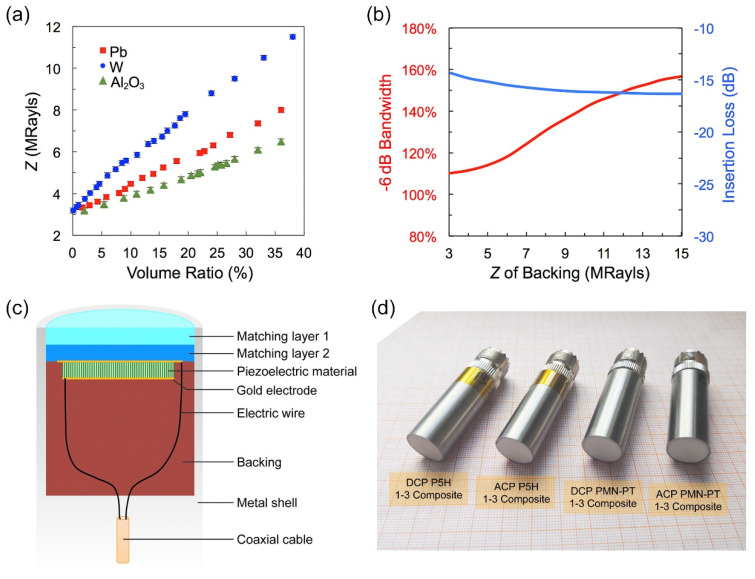
(**a**) The acoustic impedance as a function of metal-powders volume ratio, (**b**) transducer bandwidth and insertion loss as a function of metal-powders volume ratio, (**c**) designed transducer structure, (**d**) the photograph of fabricated transducers.

**Figure 5 micromachines-13-01715-f005:**
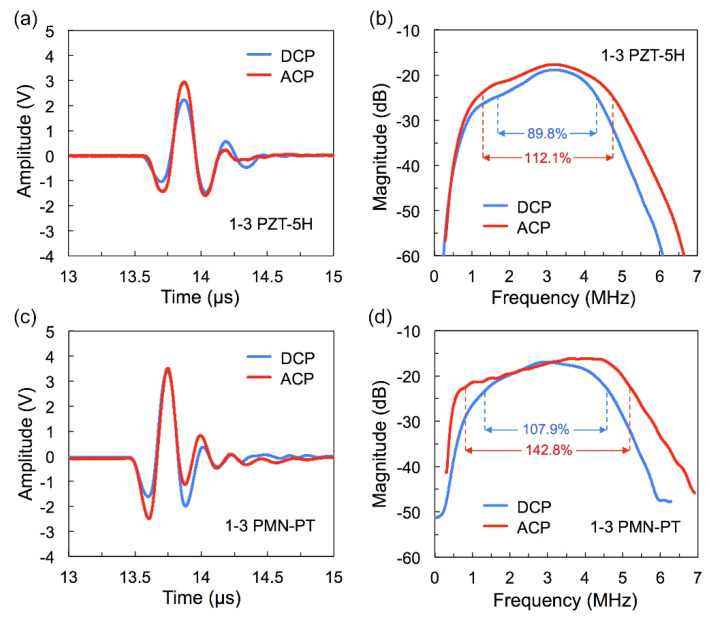
(**a**) Measured pulse-echo signals of 1–3 PZT-5H ceramic transducers, (**b**) FFT spectrum of spectrums of 1–3 PZT-5H ceramic transducers, (**c**) measured pulse-echo signals of 1–3 PMN-PT SC transducers (**d**) FFT spectrum of 1–3 PMN-PT SC transducers. (SC: single crystal).

**Figure 6 micromachines-13-01715-f006:**
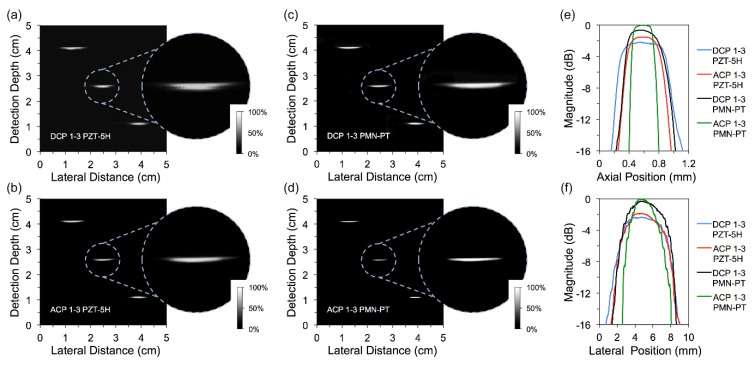
Wire box scan images of transducers with (**a**) DCP 1-3 PZT-5H ceramic, (**b**) ACP PZT-5H ceramic, (**c**) DCP 1-3 PMN-PT single crystal and (**d**) ACP 1-3 PMN-PT single crystal. The energy intensity distributions at various (**e**) axial and (**f**) lateral positions.

**Table 1 micromachines-13-01715-t001:** Optimal ACP conditions of 1–3 piezocomposites.

Material	*E* _p_	*C*	*f*
PZT-based	2 kV/mm	16	1.25 Hz
PMN-PT-based	0.9 kV/mm	10	10 Hz

**Table 2 micromachines-13-01715-t002:** Performances of the 1–3 piezocomposites under various poling methods.

Polarization	*d*_33_ (pC/N)	$\varepsilon_{33}^{T}/\varepsilon_{0}$	*k* _t_
DCP 1–3 PZT-5H	510	2480	0.66
ACP 1–3 PZT-5H	661	2896	0.75
DCP 1–3 PMN-PT	1238	1145	0.73
ACP 1–3 PMN-PT	1412	1358	0.88

**Table 3 micromachines-13-01715-t003:** Physical dimension and acoustics property parameters of each part of the transducer.

Material	Thickness (mm)	Diameter (mm)	Longitudinal Wave Velocity (m/s)	Acoustic Impedance (MRayls)	Acoustic Attenuation Coefficient (@ 3 MHz)
DCP 1–3 P5H	0.63	10	3828	22.3	-
ACP 1–3 P5H	0.63	10	3855	22.4	-
DCP 1–3 PMN-PT	0.63	10	3912	22.4	-
ACP 1–3 PMN-PT	0.63	10	3930	22.5	-
Backing(W + Epoxy)	30	11	1758	11.5	~12 dB/cm
Matching layer 1 (Al_2_O_3_ + Epoxy)	0.18	15	2144	3.8	~1 dB/cm
Matching layer 2 (Al_2_O_3_ + Epoxy)	0.27	15	3245	6.5	~1 dB/cm

**Table 4 micromachines-13-01715-t004:** The technical parameters of fabricated transducers by DCP and ACP methods.

Performance	Transducer with DCP 1–3 P5H	Transducer with ACP 1–3 P5H	Transducer with DCP 1–3 PMN-PT	Transducer with ACP 1–3 PMN-PT
Center frequency (MHz)	3.0	3.1	3.0	3.1
Insertion loss (dB)	−18.8	−17.6	−17.0	−16.1
−6 dB bandwidth	89.8%	112.1%	107.9%	142.8%

**Table 5 micromachines-13-01715-t005:** The axial and lateral −6 dB resolution of each transducer.

Performance	Transducer with DCP 1–3 P5H	Transducer with ACP 1–3 P5H	Transducer with DCP 1–3 PMN-PT	Transducer with ACP 1–3 PMN-PT
−6 dB axial resolution	0.71 mm	0.56 mm	0.59 mm	0.33 mm
−6 dB lateral resolution	6.29 mm	5.92 mm	5.79 mm	4.08 mm

## Data Availability

The data presented in this study are available on request from the corresponding author.

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
