# Peer review of "Increasing Performances of 1–3 Piezocomposite Ultrasonic Transducer by Alternating Current Poling Method"

_micromachines, 2022, doi:10.3390/mi13101715_

Round 1

Reviewer 1 Report

Authors described the influence of poling techniques on piezoelectric characteristics of the material. It seems that the same material poled under different conditions. The proposed technique was previously investigated and literature reported different works. It is thus necessary to evidence what is novel and what is not. Moreover, more analysis will be required to give a more clear idea about the effect of the proposed approach.

Abstract is poorly focused. References should be improved (most of them are a little bit outdated)

Please specify if the sample obtained were previously poled or if they have an inherent polarization prior to the ACP was applied.

Since authors used an electric characterization (Agilent 4294A multi-frequency LCR meter), impedance analysis should be provided to evidence how the resonances (in terms of R,L and C) change in the various situation.

Another point to understand is how is possible to compare the two poling techniques…Is it possible to obtain better performances using DCP or not? And Also, Can ACP be improved or the conditions were previously optimized.

Did the authors per formed P-E Hysteresis loop in both cases? I think it should be provided.

Author Response

Dear Editor and Reviewers, We really appreciate your valuable advice and also the suggestions from reviewers, which have greatly helped us improve the overall quality of the manuscript. The revised version is now resubmitted to Micromachines for your consideration. The responses to reviewer point to point are listed in the attachment, shown in italic red and color fonts. We are looking forward to your comment and decision soon. Yours sincerely, Rui Zhang September 18, 2022

Reviewer 2 Report

1.    “As an effective performance optimization method, alternating current poling (ACP) has been widely authenticated in ferroelectric materials.” Please show the reference.

2.    “Based on the systematic study, the optimal ACP conditions, including poling electric field amplitude (Ep), poling cycle number (C) and poling frequency (f) of 1-3 piezocomposites were determined, as summarized in Table 1.” How to determin the poling conditon for different piezoelectric materilas?   

3.    Please compare the properties of the PZT ceramics, PMNPT crystals, PZT 1-3 composites and PMNPT 1-3 composites, respectively.

4.    Please give the properties of the PZT and PMNPT 1-3 piezoelectric composites, such as coercive field (Ec), frequency constant, and the area of the piezoelectric composite for ultrasound transducer?  

5.    In figure 5, the reference for the different kinds of transducer performance were cited Ref.5 which was wrong. Please show the correct references. Besides, the values of the bandwith and insertion loss were not shown clearly in figure 5, please mark it.   

6.    It is better to show the test parmeters for the pulse-echo measurement, such as the gain or attenutation, the energy input and the frequency etc.

7.    Page4  line 134,  Figure 2c should be changed as Figure 1c.

8.    Does the ACP method work out on the high frequency (more than 10MHz) transducers effectively?

9.  How to fabricate the 1-3 piezoelectric composite ?

10. In figure 2, it seems cofunsed for the three kinds of materis (Pb, W and Al2O3) investigation. Which is selected for the transducers fabrication. It is better please show the specific component of the backing, matching layer 1 and matching layer 2 in table 3.

Author Response

Dear Reviewer, We really appreciate your valuable advice and also the suggestions from reviewers, which have greatly helped us improve the overall quality of the manuscript. The revised version is now resubmitted to Micromachines for your consideration. The responses to reviewer point to point are listed in the attachment, shown in italic red and color fonts. We are looking forward to your comment and decision soon. Yours sincerely, Rui Zhang September 18, 2022

Reviewer 3 Report

The authors describe an ACP poling method that improve the performance of the ultrasonic transducers. Though an interesting work, there are still two major concerns that should be addressed for a second review.

1. How about the depoling performance of piezoelectric with ACP poling method (Hysteresis loop, coercivity)?

2. For Figure 4, quantitive analysis is necessary (SNR, lateral resolution with -6dB).

Author Response

(The authors gave the same response as above.)

Round 2

Reviewer 1 Report

Authors have addressed most of the comments. Below my remaining questions:

from P-E hysteresis loop, it seems that the remanent polarization is quite similar in all cases. A more in depth discussion is thus needed. Is the improvement statistically significant for example respect to DCP-1-3-PMN-PT. Are there variability in the data on multiple samples? 

Impedance anaysis can be plotted in ohm rather than in dB. Also, a discussion about the impedance data should be provided.

Author Response

Dear Editor and Reviewers, We really appreciate your valuable advice and also the suggestions from reviewers, which have greatly helped us improve the overall quality of the manuscript. The revised version is now resubmitted to Micromachines for your consideration. The response to reviewer point to point are listed in the attachment, shown in red color. Some necessary references have been added. We are looking forward to your comment and decision soon. Yours sincerely, Rui Zhang September 21, 2022

Reviewer 2 Report

accept

Author Response

Dear Editor and Reviewers, We really appreciate your previous valuable advice and suggestions from reviewers, which have greatly helped us improved the overall quality of the manuscript. The revised version is resubmitted to Micromachines for your consideration. We really hope to have more and better research results to share in the future. Yours sincerely, Rui Zhang September 21, 2022

Reviewer 3 Report

The author addressed the all concerns. This manuscript can be published in present form.

Author Response

(The authors gave the same response as above.)
